# Path analysis of factors associated with nurses' pain management practices in older adults with cognitive impairment: A cross-sectional study

Phichpraorn Youngcharoen[1], Atsadaporn Niyomyart[1*], Piyawadee Thongyost[1], Joachim G. Voss[2]

**1** Ramathibodi School of Nursing, Faculty of Medicine Ramathibodi Hospital, Mahidol University, Khwaeng Thung Phaya Thai, Khet Ratchathewi, Bangkok, Thailand, **2** University of Nebraska Medical Center (UNMC), College of Nursing - Omaha Division, Nebraska Medical Center, Omaha, Nebraska, United States of America

* atsadaporn.niy@mahidol.ac.th

## Abstract

### Background

Pain management is essential, yet inadequate management is linked to anxiety, depression, and poor quality of life. Evidence in Thailand is limited for older adults with cognitive impairment. This study examined factors associated with pain management practices among nurses.

### Design

Secondary descriptive correlational study.

### Methods

A secondary cross-sectional analysis used an existing dataset (1 September-27 October 2023); no new data were collected. Institutional Review Board approval was obtained on 4 February 2024; the dataset was accessed on 5 February 2024. Guided by Social Cognitive Theory, 174 full-time registered nurses completed self-administered paper questionnaires, including a modified version of the Tool for Evaluating the Ways Nurses Assess Pain, the Collaboration and Satisfaction Care Decisions Instrument, and the Pain Management Self-Efficacy Questionnaire. Data were analyzed using descriptive statistics, Spearman's correlation, and structural equation modeling.

### Results

All nurses were female; most held a bachelor's degree (95.40%); mean age 31.47±6.98 years. The model showed good fit and explained 37% of the variance in nurses' pain management practices. Direct effects on nurses' pain management practices were observed for nurses' perceptions of collaboration with physicians

**Data availability statement:** The code-book supporting the findings of this study is available as Supporting Information. The anonymized dataset is available from the corresponding author upon reasonable request.

**Funding:** Ramathibodi School of Nursing, Faculty of Medicine Ramathibodi Hospital, Mahidol University, will support the publication fee.

**Competing interests:** The authors have declared that no completing interests exist.

(β = 0.28, p < 0.001, 95% CI [0.16, 0.41]) and nurses' pain management self-efficacy (β = 0.34, $p$ < 0.001, 95% CI [0.20,0.47]). Nurses' knowledge and attitudes toward pain management, nurses' perceptions of collaboration with physicians, and years of nursing experience also had indirect effects on nurses' pain management practices through nurses' pain management self-efficacy (β = 0.08, p < 0.05, 95% CI [0.02, 0.14]), β = 0.08, p < 0.01, 95% CI [0.03, 0.13], and β = 0.13, p < 0.01, 95% CI [0.05, 0.20], respectively.

## Conclusions

Pain management self-efficacy plays a key role in nursing practice. Building it through targeted interventions, training, and institutional support may improve pain management competencies for older adults with cognitive impairment.

## 1. Background

Pain in older adults with cognitive impairment requires appropriate assessment and management. Approximately 50% to 66% of individuals experience pain, with the most common pain-related conditions are knee osteoarthritis (29.27%), headache (12.53%), and osteoporosis (11.43%) [1,2]. Among older adults with cognitive impairment, such as those diagnosed with dementia, about 48% experience moderate to severe pain, which is associated with increased agitation, functional decline, resistance to care, and further cognitive deterioration [1,3,4]. When inadequately managed, pain may contribute to psychological problems such as anxiety and depression, thereby negatively affecting overall quality of life [5]. Therefore, effective pain management practices among nurses caring for individuals with cognitive impairment are essential for alleviating pain and improving patient outcomes.

Nursing practices play a pivotal role in appropriate pain management. However, older adults with cognitive impairment frequently experience under-assessment and delayed treatment, resulting in suboptimal pain outcomes [6,7]. Integrative and systematic reviews indicate that nurses often lack essential pain assessment skills and have limited knowledge of appropriate assessment tools [6,8]. In addition, misconceptions that pain is a normal part of aging, the absence of standardized pain assessment practices in clinical settings, and communication limitations among individual adults with cognitive impairment further complicate effective pain management [9–11]. For example, although 88% of older adults with dementia are assessed for pain following surgery, only one-third receive analgesic treatment compared with other postoperative patients [12]. This discrepancy underscores a critical gap between pain recognition, knowledge, and appropriate pain management in this population.

Knowledge and attitudes toward pain management, years of nursing experience, and perceptions of collaboration with physicians among nurses have been associated with effective pain management practices [13–16]. Previous studies indicate that nurses with greater clinical experience demonstrate more effective pain management

practices, and that their perceptions of collaboration with physicians are associated with pain management practices among older adults [13–15]. Beyond clinical experience and collaborative factors, knowledge and attitudes toward pain management have also been identified as important determinants of self-efficacy in pain management practices [17]. In the context of pain management, self-efficacy reflects nurses' perceptions of their ability to assess and manage pain, thereby enhancing the quality of care and promoting patient safety [18].

Self-efficacy has been found to serve as a mediator between knowledge and attitudes toward pain management and actual pain management practices among nurses [17]. However, despite its importance, a lack of confidence among nurses in managing pain medications has been reported [17]. Given pain management practices remain understudied and that factors associated with these practices require further investigation, this study specifically aims to examine the relationships among knowledge and attitudes toward pain management, perceptions of collaboration with physicians, years of nursing experience, pain management self-efficacy, and pain management practices among nurses.

### 1.1 Conceptual framework

This study was guided by Social Cognitive Theory (SCT) and relevant literature. Social Cognitive Theory, developed by Albert Bandura, describes how personal, behavioral, and environmental factors interact to influence individual behavior [19]. Although SCT includes multiple key components (i.e., self-efficacy, outcome expectations, observational learning, and reinforcement), this study focused on those most directly related to nurses' pain management practices in clinical settings [19]. Thus, only theoretically relevant components were operationalized.

Fig 1 presents the study's theoretical framework based on the concept of triadic reciprocal determinism, which conceptualizes behavior as the result of interactions among personal, behavioral, and environmental factors. In this study, self-efficacy, defined as an individual's belief in their ability to perform specific tasks, is conceptualized as a central personal (cognitive) factor [19]. Nurses' knowledge and attitudes toward pain management are also considered personal cognitive determinants that contribute to the development of self-efficacy [19,20]. Years of nursing experience are conceptualized as a source of mastery experience that may strengthen self-efficacy [15,16]. Nurses' perceptions of collaboration with physicians are considered environmental influences, while pain management practices represent behavioral outcomes. Together, these factors have been associated with self-efficacy and pain management practices [15,16].

Drawing on SCT and prior literature, this study hypothesizes that nurses' knowledge and attitudes toward pain management, nurses' perceptions of collaboration with physicians, and years of nursing experience influence nurses' pain management practices both directly and indirectly through nurses' pain management self-efficacy. Self-efficacy is proposed as a mediating variable linking personal and environmental factors to behavioral outcomes; individuals with higher

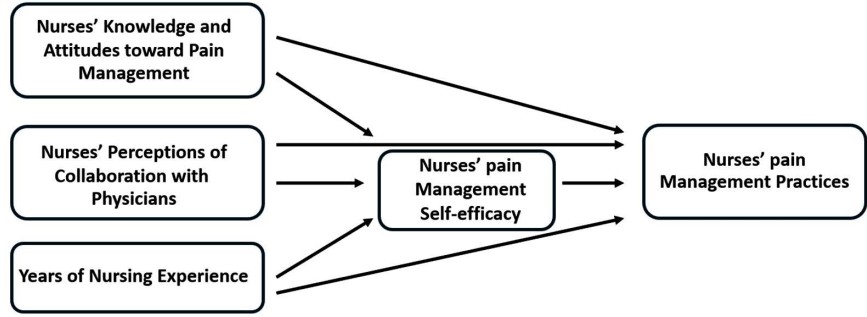

**Fig 1. The hypothesized model for nurses' pain management practices.**

self-efficacy are more likely to engage in and perform desired behaviors [19,21]. See **Fig 1** for the study's theoretical framework details.

## 2. Methods

### 2.1 Study design

This study is a secondary data analysis using data from a cross-sectional study (COA No. MURA2023/290 and IRB No.026/2566) conducted between 1 September and 27 October 2023. Approval from the IRB to use the secondary dataset was obtained on 4 February 2024. The dataset was accessed on 5 February 2024 for analysis. The secondary analysis was aligned with both the study objectives and the original purpose of data collection, supported by validated measurement tools and an adequate sample size. The STROBE checklist was followed [22].

### 2.2 Settings

The original study used a paper-based self-administered questionnaire to evaluate nurses' knowledge and attitudes toward pain management, nurses' pain management self-efficacy, nurses' collaboration with physicians, work complexity, and nurses' pain management practices for older adults with cognitive impairment at a quaternary university hospital and a tertiary care hospital in central Thailand. These two hospitals, overseen by the Thai Red Cross and the Ministry of Higher Education, Science, Research, and Innovation, were selected randomly. Both hospitals accept referred patients from primary and secondary care hospitals and provide specialized care for patients with complex conditions, utilizing highly specialized equipment and expertise. These hospitals also implemented frontline cognitive assessment tools, using the Mini-Mental State Examination (MMSE) or orientation to time, place, and person to screen cognitive impairment in older adults prior to providing treatment and nursing care.

### 2.3 Participants

This study used data from a previously conducted study (unpublished). The original study recruited 180 registered nurses from two hospitals using a convenience sampling method, achieving a response rate of 90%. Among the 20 non-respondents, 5 explicitly declined participation due to time constraints, while 15 did not return the survey by the closing date.

After excluding six outlier cases, a total of 174 nurses were included in the final analysis. Participants met the following inclusion criteria: (1) full-time registered nurses working in adult or older adult inpatient units for at least 20 hours per week; (2) at least one year of experience providing care to older adults with cognitive impairment; and (3) willingness to participate in the study. Exclusion criteria included advanced practice nurses, nurses in administrative roles (i.e., head nurses, nurse supervisors), and those who did not provide consent.

The prior sample size for this secondary data analysis was estimated using G*Power software (version 3.1.9.7) based on a multiple linear regression model representing the most complex path in the proposed model. Assuming a medium effect size of 0.15, a statistical power of 0.80, an alpha level of 0.05, and four predictors, the required sample size was 85. To ensure adequate power for the structural equation modeling (SEM) analysis, the sample size was further evaluated based on recommended criteria for SEM. A sample size of 10–20 participants per estimated parameter was considered appropriate [23]. Given that the model included 12 parameters, the recommended sample size ranged from 120 to 240 participants. The final sample size of 174 participants in this study fells within this recommended range, indicating that the study was adequately powered to detect statistically significant relationships.

### 2.4 Data collection

Based on the original study, data were collected from nurses in regular and critical care units through a paper-based self-questionnaire. The principal investigator (PI) initially contacted head nurses across units to obtain permission and subsequently introduced the study to eligible nurses.

Four research assistant nurses completed a 2-hour onsite training session for data collection procedures, during which they had opportunities to ask questions and receive feedback from the PI prior to data collection. Participants were not compensated for their participation.

After obtaining permission to present the study during monthly unit meeting, a research assistant explained the study's purpose, estimated completion time, and provided contact details. Participants were informed that participation was voluntary and anonymous, and that their decision would not affect their performance evaluations. Written informed consent was obtained after participants reviewed an information sheet outlining the study purpose, risks, and benefits. Participants who consented completed the questionnaire privately, which required approximately 30–40 minutes. Those who declined participation were free to leave without any consequences.

Upon return of each questionnaire, a research assistant performed a brief completeness check. Participants were notified of any accidentally missed items and were given the option to complete them or submit the questionnaire without modification. Initial screening identified minimal missing data, including one missing item in the knowledge and attitudes scale (n = 2; 0.09%), one missing item in the self-efficacy scale (n = 2; 0.05%), and one missing item in the pain management practices scale (n = 1; 0.02%). All participants chose to complete the missing items, resulting in a final dataset with no missing data.

All questionnaires were collected anonymously, and only the PI had access to the data. The data were securely stored, and all results were reported without identifiable information.

## 2.5  Instruments

All study tools presented in this study were based on the unpublished study protocol COA. MURA2023/290 and IRB No. 026/2566.

**2.5.1  Demographic information.**  Demographic information of participants included age, gender, the highest education level, workplace, working unit, years of nursing experience in total and providing care for older adults with cognitive impairment, pain management training in older adults with cognitive impairment, and resources to improve pain management knowledge were also collected.

**2.5.2  Tool for Evaluating the Ways Nurses Assess Pain (TENAP).**  Nurses' knowledge and attitudes toward pain management and pain management practices in older adults with cognitive impairment were measured using the modified version of the Tool for Evaluating the Ways Nurses Assess Pain (TENAP), developed by Ng et al., 2014 [24]. After receiving permission, the tool was translated from English to Thai using forward-backward translation, and its face and content validity were evaluated by a panel of five healthcare professionals. The TENAP consists of two parts.

Part A consisted of 13 items to assess nurses' knowledge and attitudes toward pain management in older adults with cognitive impairment. Participants were asked to select response as true, false, or "do not know". Responses were scored as 1 for correct answers and 0 for incorrect or "do not know" responses, with total scores ranging from 0 to 13. A higher score indicated higher knowledge and positive attitudes toward pain management. The scale content validity index (S-CVI) of TENAP (Part A) was 0.84. The construction validity was confirmed through exploratory factor analysis, with one factor explaining 15.40% of the variance. The Cronbach's alpha coefficient was 0.75.

Part B assessed nurses' practices related to pain assessment in older adults with cognitive impairment. It included two case-study vignettes: one describing a patient with post-stroke who experienced pain (vignette 1) and another describing a patient with fever of unknown origin and mild confusion (vignette 2). There were six pain assessment behavior items for each vignette (total 12 items across two vignettes). Responses were rated on a 5-point Likert scale ranging from 0 (unsure) to 4 (on every occasion that care was provided). The total score ranged from 0 to 24 for each case study vignette. A higher score indicated higher pain assessment practices in this population. The CVI of TENAP (Part B) was 1.00. The construct validity was confirmed through exploratory factor analysis. The Cronbach alpha's coefficients of case study vignettes 1 and 2 were 0.87 and 0.90, respectively.

In this study, we revised the response options for TENAP (Part A)-Thai version after receiving permission from the developer, including 13 items to true (1 point) and false or "do not know" (0 points). As no existing instrument specifically evaluates pain management in older adults with cognitive impairment, we reviewed the literature and obtained approval from the developer to expand the instrument. Thus, 11 additional items were added to each vignette to assess nurses' pain management practices in this population. Following the development, the Thai questions were submitted to five Thai experts for feedback on clarity, relevance, and logical flow. They suggested minor changes for better understanding. The added items did not require to be returned to the developer, as the developer's focus was on pain assessment rather than management. Here are examples of the questions added "I will re-assess pain after administering oral and intravenous analgesics within 30 minutes and 60 minutes, respectively" [25]. After administering opioid analgesics, I will assess the sedative score and related side effects [26]. Thus, part B of the TENAP-Thai version consisted of 12 original items assessing pain assessment behaviors and 22 newly developed items assessing pain management behaviors. Based on the Part B scoring scale, with item scores ranging from 0 to 4, the total score was calculated by summing all 34 items, yielding a possible range of 0–136, with higher scores indicating better pain assessment and management practices. Given this expanded scoring structure, the present study represents the Thai adaptation and expansion of the instrument to include additional items measuring pain management practices, defined as the implementation of pharmacological and non-pharmacological approaches to alleviate pain [27]. In contrast, the original TNAP assessed nurses' knowledge and attitudes toward pain management, as well as pain assessment focused on the evaluation of pain. These differences limit direct comparison between the original and modified TENAP.

The TENAP parts A and B were translated into Thai using a back-translation procedure. First, a bilingual nursing instructor fluent in both Thai and English translated the instrument from English into Thai. A second nursing instructor independently back translated the Thai version into English. The back-translated version was then reviewed by a native English-speaking expert to ensure consistency with the original TENAP. Five experts, including a geriatrician with expertise in cognitive function, a registered nurse from an acute pain service, two advanced practice nurses specializing in dementia care for older adults and the other in gerontology with expertise in pain management, and a clinical pharmacist reviewed the questionnaire items for clarity and content relevance. For part A, the S-CVI and I-CVI were 0.85 and 0.97, respectively. For part B, the S-CVI and I-CVI were both 1. The Thai version of TENAP was pilot-tested with 20 registered nurses who had the same inclusion criteria as participants in this study, and the Kuder-Richardson 20 for internal reliability of the TENAP-Thai version part A was 0.83. For part B, the Cronbach's alpha coefficients of the six items assessing pain assessment for case study vignettes 1 and 2 were 0.79 and 0.90, respectively. The Cronbach's alpha coefficients of the 11 items measuring pain management for case study vignettes 1 and 2 were 0.73 and 0.79, respectively. In this study, the Kuder-Richardson 20 of the TENAP-Thai version with 174 nurses in part A was 0.67.

For part B, exploratory factor analysis using the principal component method was performed to verify the construct validity of the newly developed items. Factor loadings for the 11 items measuring pain management ranged from 0.61 to 0.78 for case study vignette 1 and from 0.61 to 0.84 for case study vignettes 2. The Kaiser-Meyer-Olkin (KMO) for the case study vignette 1 was 0.87, which is acceptable. Bartlett's test of sphericity was also significant ($x2 = 1025.217$, $df = 55$, $p < 0.001$). For case study vignette 2, the KMO was 0.89, and Bartlett's test of sphericity was also significant ($x2 = 1250.711$, $df = 55$, $p < 0.001$). To further examine construct validity, convergent and divergent validity were assessed. A moderate positive correlation was observed between the newly developed pain management practice items and pain management self-efficacy ($r_s = 0.43$, $p < 0.001$), supporting convergent validity. In contrast, no significant correlation was found between the new pain management practice items and work complexity ($r_s = -0.06$, $p = 0.437$), providing evidence of divergent validity.

Regarding internal consistency reliability, the Cronbach's alpha coefficients for the 6-item pain assessment scale for case study vignettes 1 and 2 were 0.87 and 0.85, respectively. For the 11-item pain management scale, Cronbach's alpha coefficients were 0.90 for vignette 1 and 0.91 for vignette 2, indicating high reliability.

### 2.5.3 Collaboration and Satisfaction about Care Decisions Instrument (CSACD).

Nurses' perceptions of collaboration with physicians in patient care were measured using the 9-item Collaboration and Satisfaction about Care Decisions Instrument (CSACD) [28]. Responses were rated on a 7-point Likert scale ranging from 1 (strongly disagree/ not satisfied/no collaboration) to 7 (strongly agree/very satisfied/complete collaboration). Total scores range from 9 to 63, with higher scores indicating strong perceived collaboration with physicians in patient care. The construct validity of the CSACD was confirmed through exploratory factor analysis, demonstrating good convergent validity. Internal consistency reliability for the six items measuring critical attribute of collaboration was 0.93, and the correlation coefficient between the two satisfaction items was 0.64 [28]. The instrument was translated from English to Thai using a committee translation approach, along with cognitive interviews. The I-CVI and the S-CVI for the Thai version, as assessed by five experts, were 0.97 and 0.88, respectively. A previous study reported a Cronbach's alpha coefficient of 0.70 for the Thai version [16], whereas the Cronbach's alpha in the present study was 0.90 among 174 nurses.

### 2.5.4 Pain Management Self-Efficacy Questionnaire (PMSEQ).

Nurses' pain management self-efficacy was measured using the 21-item Pain Management Self-Efficacy Questionnaire (PMSEQ) [18]. Response options ranged from 0 (not confident at all) to 5 (highly confident), with total scores ranging from 0 to 105. Higher scores indicate a higher perception of pain management self-efficacy. The Cronbach's alpha coefficient for the original instrument was 0.96. Construct validity was confirmed through exploratory factor analysis in which comprehensive, evaluative, and supplemental pain management self-efficacy accounted for 39.15%, 20.57%, and 15.65% of the total variance, respectively [18]. The instrument was translated into Thai for this study, with the I-CVI and the S-CVI of 1.00 each. A previous study reported a Cronbach's alpha coefficient of 0.93 for the Thai version [29], whereas the Cronbach's alpha in the present study was 0.97 among 174 nurses.

## 2.6 Ethical consideration

The ethical approval for this secondary data study was granted by the Committee on Human Rights Related to Research Involving Human Subjects, Faculty of Medicine, Ramathibodi Hospital, Mahidol University IRB (COA. MURA2024/97), approved on February 4, 2024.

## 2.7 Data analysis

Data were analyzed using Stata version 18. Descriptive statistics were used to describe demographic information and study variables, including mean, standard deviation, median, frequency, and percentage.

Variance inflation factors (VIFs), skewness, and kurtosis were assessed to evaluate potential violations of multicollinearity and normality assumptions. In this study, all VIF values were below 10, indicating no evidence of problematic multicollinearity among the independent variables. However, the distributions of years of nursing experience and nurses' pain management practice scores demonstrated notable skewness, suggesting deviations from normality.

Spearman's correlation coefficient was used to assess the relationships among all study key variables.

The path analysis in SEM was established due to its ability to examine both direct and indirect effects among multiple independent and dependent variables simultaneously [30]. In addition, SEM is well suited for evaluating mediation processes that involve multiple independent variables, mediators, and outcomes within a single analysis [30]. In this study, path analyses were conducted using SEM in Stata to identify the relationships among nurses' knowledge and attitudes toward pain management, nurses' perceptions of collaboration with physicians, years of nursing experience, nurses' pain management self-efficacy, and nurses' pain management practices in older adults with cognitive impairment. To estimate direct effects and test mediation effects, a bootstrapping method with 5,000 resamples was employed. Bootstrapping is recommended for evaluating the significance of indirect effects as it does not rely on the assumption of normality of the sampling distribution and provides more accurate confidence intervals, thereby reducing the risk of Type I error [31].

The model was estimated using the maximum likelihood method. We also examined the goodness of fit index following the indices of model fit, the Tucker-Lewis index (TLI > 0.90), the comparative fit index (CFI > 0.90), the standardized root mean square residual (SRMR < 0.08, indicates good fit), and the root mean square error of approximation (RMSEA < 0.08) [32].

## 3 Results

### 3.1 Sample characteristics

The characteristics of study participants are presented in Table 1. The mean age of participants was 31.47 years (SD = 6.98), and the median years of nursing experience was 8 (IQR = 3–13), ranging from 1 to 29 years. The sample (N = 174) was primarily female nurses (100%; n = 174) working in regular (81.61%; n = 142) and critical care (18.39%; n = 32) units. Most participants held a baccalaureate degree in nursing (95.40%; n = 166). A significant portion of the participants had not previously received training in pain management (83.33%; n = 145), pain management for older adults

**Table 1. Participant characteristics (N = 174).**

| Variable | N (%) |
|---|---|
| **Age:** Min = 22, Max = 51, Mean = 31.47, SD = 6.98 | |
| **Years of nursing experience:** Min = 1, Max = 29, Median = 8, IQR (3 –13) | |
| < 5 years | 58 (33.33) |
| 5-10 years | 57 (32.76) |
| > 10 years | 59 (33.91) |
| **Gender** | |
| Male | 0 (0.00) |
| Female | 174 (100.00) |
| **Education level** | |
| BSN | 166 (95.40) |
| Master's degree | 8 (4.60) |
| **Working unit** | |
| Regular | 142 (81.61) |
| Critical care | 32 (18.39) |
| **Pain management training in the past year** | |
| Yes | 29 (16.67) |
| No | 145 (83.33) |
| **Pain management training for older adults in the past year** | |
| Yes | 18 (10.34) |
| No | 156 (89.66) |
| **Pain management training for older adults with cognitive impairment in the past year** | |
| Yes | 10 (5.75) |
| No | 164 (94.25) |

The mean scores were as follows: nurses' knowledge and attitudes toward pain management (M = 5.61, SD ± 2.63), nurses' perceptions of collaboration with physicians (M = 52.72, SD ± 6.88), nurses' pain management self-efficacy (M = 85.43, SD ± 12.90), and nurses' pain management practices (M = 117.57, SD ± 13.49). See **Table 2** for details.

**Table 2. Descriptive statistics of Nurses' knowledge and attitudes toward pain management, nurses' perceptions of collaboration with physicians, nurses' pain management self-efficacy, nurses' pain management practices (N = 174).**

| Study variables | Possible range | Actual range | Mean | S.D. | Median (IQR) | Interpretation |
|---|---|---|---|---|---|---|
| Nurses' knowledge and attitudes toward pain management | 0-13 | 1-13 | 5.61 | 2.63 | | Moderate |
| Nurses' perceptions of collaboration with physicians | 9-63 | 26-63 | 52.72 | 6.88 | | High |
| Nurses' pain management self-efficacy | 0-105 | 58-105 | 85.43 | 12.90 | | High |
| Nurses' pain management practices | 0-136 | 68-136 | 117.57 | 13.49 | 121 (106-130) | High |

**Footnote.** Nurses' knowledge and attitudes toward pain management: higher scores indicate higher knowledge and positive attitudes toward pain. Nurses' perceptions of collaboration with physicians: higher scores indicate strong perceived collaboration with physicians in patient care. Nurses' pain management self-efficacy: higher scores indicate a higher perception of pain management self-efficacy. Nurses' pain management practices: higher scores indicating better pain assessment and management practices.

(89.66%; n = 156), or pain management for older adults with cognitive impairment (94.25%; n = 164) in the past year. See **Table 1** for details.

**Table 3** presents the Spearman's rank-order correlations and descriptive statistics of the study variables. Nurses' pain management self-efficacy showed moderate positive correlations with years of nursing experience ($r_s = 0.45$, $p < 0.001$) and nurses' knowledge and attitudes toward pain management ($r_s = 0.32$, $p < 0.001$), and a weak correlation with nurses' perceptions of collaboration with physicians ($r_s = 0.21$, $p < 0.01$). Nurses' pain management practices showed moderate correlations with nurses' perceptions of collaboration with physicians ($r_s = 0.36$, $p < 0.001$) and nurses' pain management self-efficacy ($r_s = 0.39$, $p < 0.001$), and a weak correlation with years of nursing experience ($r_s = 0.26$, $p < 0.001$). Nurses' knowledge and attitudes toward pain management were not significantly correlated with nurses' pain management practices ($r_s = 0.03$, $p > 0.05$). See **Table 3** for details.

### 3.2 Testing the model of nurses' pain management practices in older adults with cognitive impairment

Because the hypothesized model was a just-identified model with zero degrees of freedom, the fit indices could not be evaluated. The initial model was refined by removing this path from years of nursing experience to nurses' pain management practices to achieve parsimony while maintaining alignment with theory and previous empirical evidence. Within SCT, years of nursing experience is conceptualized as contributing to behaviors through nurses' management self-efficacy

**Table 3. Spearman's correlations and descriptive statistics of study variables (N = 174).**

| Variable | 1 | 2 | 3 | 4 | 5 |
|---|---|---|---|---|---|
| 1. Nurses' knowledge and attitudes toward pain management | – | | | | |
| 2. Nurses' perceptions of collaboration with a physician | −0.15* | – | | | |
| 3. Years of nursing experience | 0.37*** | − 0.01 | – | | |
| 4. Nurses' pain management self-efficacy | 0.32*** | 0.21** | 0.45*** | – | |
| 5. Nurses' pain management Practices | 0.03 | 0.36*** | 0.26*** | 0.39*** | – |

* $p < 0.05$ **; $p < 0.01$; ***$p < 0.001$

rather than as a direct predictor [16,17,33]. As shown in **Fig 2**, the final model showed adequate fit to the data: $\chi^2 = 1.642$ (*df* = 1, *p* = 0.200), TLI = 0.956, CFI = 0.994, RMSEA = 0.061, and SRMR = 0.019. The final model accounted for 37% of the variance in nurses' pain management practices ($R^2 = 0.37$), indicating that the model explains a moderate amount of variance. The remaining 63% suggests that additional factors not included in the model may influence pain management practices.

### 3.3 Association among nurses' knowledge and attitudes toward pain management, nurses' perceptions of collaboration with physicians, years of nursing experience, and nurses' pain management self-efficacy

Nurses' knowledge and attitudes toward pain management (β = 0.23, *p* < 0.01, 95% CI [0.10, 0.36]) and nurses' perceptions of collaboration with physicians (β = 0.24, *p* < 0.001, 95% CI [0.12, 0.36]) had significant direct positive effects on nurses' pain management self-efficacy. The standardized coefficients indicated small to moderate effects, and the 95% confidence intervals did not include zero, suggesting statistically reliable estimates consistent with the hypothesized model.

Years of nursing experience had a significant direct positive effect on pain management self-efficacy (β = 0.37, *p* < 0.001, 95% CI [0.25, 0.50]), indicating a moderate to large effect, with the confidence interval entirely above zero.

### 3.4 Association among nurses' knowledge and attitudes toward pain management, nurses' perceptions of collaboration with physicians, years of nursing experience, nurses' pain management self-efficacy, and nurses' pain management practices

Nurses' pain management self-efficacy had a significant direct positive effect on nurses' pain management practices (β = 0.34, *p* < 0.001, 95% CI [0.20, 0.47]), indicating a moderate effect. Similarly, nurses' perceptions of collaboration with physicians had a significant direct positive effect on nurses' pain management practices (β = 0.28, *p* < 0.001, 95% CI [0.16, 0.41]), also reflecting a moderate effect. For both effects, the confidence intervals did not include zero, indicating statistically significant estimates. However, nurses' knowledge and attitudes toward pain management did not have a significant direct effect on nurses' pain management practices (β = −0.04, *p* = 0.594, 95% CI [−0.18, 0.11]). This suggests that knowledge and attitudes toward pain management do not directly influence practices, but rather operate indirectly through nurses' pain management self-efficacy.

The indirect effects were estimated using bootstrapping with 5,000 resamples. Nurses' pain management self-efficacy significantly mediated the relationships between nurses' knowledge and attitudes toward pain management and nurses' pain management practices (β = 0.08, *p* < 0.05, 95% CI [0.02, 0.14]) and between nurses' perceptions of collaboration

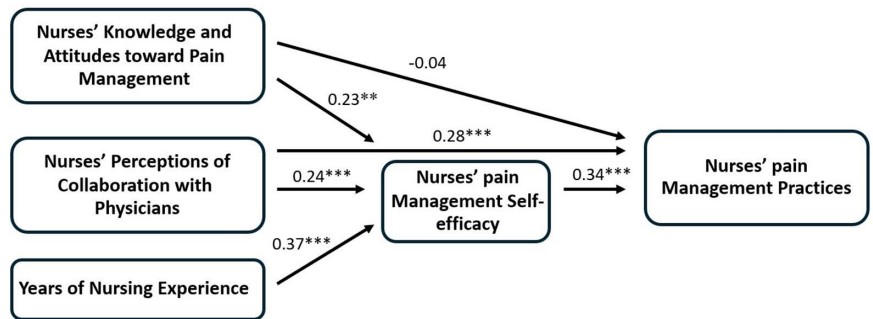

**Fig 2. A final modified model of predicting nurses' pain management practices in older adults with cognitive impairment.** Note. **\*\*\****p* < .01; \*\*\**p* < .001.

with physicians and nurses' pain management practices (β = 0.08, *p* < 0.01, 95% CI [0.03, 0.13]), indicating small indirect effects. The confidence intervals for both indirect effects did not include zero, supporting the presence of mediation.

Years of nursing experience had a significant indirect effect on nurses' pain management practices through nurses' pain management self-efficacy (β = 0.13, *p* < 0.01, 95% CI [0.05, 0.20]), indicating a moderate indirect effect. See **Table 4** for details.

## 4. Discussion

This study examined the relationships among nurses' knowledge and attitudes toward pain management, nurses' perceptions of collaboration with physicians, years of nursing experience, nurses' pain management self-efficacy, and nurses' pain management practices in older adults with cognitive impairment in Thailand.

Overall, nurses in the present study reported lower levels of knowledge and attitudes toward pain management (M = 5.61, SD ± 2.63). These findings are consistent with studies conducted in the United States, Australia, Spain, and Sweden, which have reported suboptimal levels of pain-related knowledge among nurses [34–37]. Similarly, the low attitude scores observed in this study align with findings from Vietnam and Jordan, where negative attitudes have been linked to uncertainty in pain management [38,39]. Notably, only 5.75% of nurses in the present study had received training related to pain management in older adults with cognitive impairment; therefore, these knowledge deficits and negative attitudes toward pain management are not unexpected. In addition, nurses' knowledge and attitudes toward pain management were found to positively influence nurses' pain management practices; however, this effect was indirect and mediated by nurses' pain management self-efficacy. These findings suggest that knowledge and attitudes toward pain management alone are insufficient to predict clinical practice. Instead, nurses' pain management self-efficacy functions as a key pathway through which knowledge and attitudes toward pain management are translated into clinical action [19]. Prior studies have shown that nurses with higher self-efficacy are more capable of applying their knowledge and making appropriate clinical decisions in pain management practices [40,41]. Further, regular and comprehensive training can strengthen nurses' knowledge and attitudes toward pian management as well as confidence, thereby promoting greater effort in delivering effective pain management practices [29,42–44]. These underscore the importance of professional training in enhancing nurses' confidence, particularly in the care of older adults with cognitive impairment who rely heavily on nurses' clinical judgment.

Our study showed that nurses' pain management practices were significantly associated with nurses' perceptions of collaboration with physicians. This finding is consistent with prior studies conducted in Japan, China, and Norway [45–47], which highlight the importance of nurses-physician collaboration in enhancing pain management practices. Effective collaboration has also been found to be associated with shared responsibility in patient care, facilitates nurses'

**Table 4. Standardized coefficients of all direct and indirect effects on pain management practice (N = 174).**

| Pathway | Standard-ized Coef. | p | 95% CI [Lower, upper] |
|---|---|---|---|
| Nurses' knowledge and attitudes toward pain management ◊ Nurses' pain management self-efficacy ◊ Nurses' pain management practices | 0.08 | < 0.05 | [0.02, 0.14] |
| Nurses' perceptions of collaboration with physicians ◊ Nurses' pain management self-efficacy ◊ Nurses' pain management practices | 0.08 | < 0.01 | [0.03, 0.13] |
| Years of nursing experience ◊ Nurses' pain management self-efficacy ◊ Nurses' pain management practices | 0.13 | < 0.01 | [0.05, 0.20] |

*Note.* Coef = coefficient; CI = confidence interval

confidence, and supports more active clinical decision-making, which may contribute to improve pain management practices [16,45,48]. When examined using SEM, nurses' perceptions of collaboration with physicians had a significant direct effect on nurses' pain management practices, indicating that higher levels of collaboration are associated with better pain management practices. This finding is supported by a study in China demonstrating that interprofessional team collaboration positively influences nurses' competency [49]. Previous research also suggests that interdisciplinary collaboration is associated with improved care quality and patient outcomes [50,51]. Thus, strengthening collaboration between nurses and physicians is essential, as it may facilitate a shared understanding of patients' pain experiences while ensuring timely and appropriate interventions.

Years of nursing experience demonstrated an indirect effect on pain management practices through nurses' pain management self-efficacy. This finding contrasts with studies from Japan, Eritrea, and Saudi Arabia, which reported that clinical nurses' experience was associated with pain management practices [47,52,53]. Similarly, evidence from scoping and systematic reviews suggests that clinical experience contributes to the development of nursing competencies and practice performance [54,55]. However, our results are consistent with studies conducted among Thai nurses caring for older surgical patients [16], nurses in Israel assessing pain in mechanically ventilated patients [13], and Iranian nurses using non-pharmacological pain management strategies [56], all of which reported no direct effect of experience on practices. These inconsistencies may be explained by the complex and subjective nature of pain, which involves biological, social, and psychological dimensions [57]. Years of nursing experience alone may not be sufficient to ensure effective pain management practices unless it is translated into confidence and competent action. In this context, pain management self-efficacy plays a crucial role by serving as a bridge between experience and practice. It reflects not only nurses' confidence in their ability to assess and interpret pain, but also their capability to implement appropriate pharmacological and non-pharmacological approaches [58]. Therefore, interventions should move beyond experience accumulation and focus on strengthening self -efficacy through master experiences, modeling, and verbal persuasion [19], for example, simulation, brief video-based training, and tailored feedback messages. Such approaches may facilitate the translation of nursing experience into effective pain management practices.

The SEM analysis indicated that nurses' knowledge and attitudes toward pain management, nurses' perceptions of collaboration with physicians, years of nursing experience, and nurses' pain management self-efficacy explained 37% of the variance in nurses' pain management practices. Although several significant individual-level factors were identified, a substantial proportion of variance in pain management practices remained unexplained. Previous research suggests that pain management practices is influenced not only by individual characteristics but also by organizational factors [59,60]. In particular, supportive clinical system, including structured education, professional training, and practice support, are critical for facilitating effective and sustained improvements in pain management practices [61]. Therefore, system-level factors within the Thai healthcare context should be further investigated. Elements such as clinical guidelines, institutional policies, standardized pain assessment tools, and multidisciplinary pain teams are often concentrated in urban and larger hospital settings, potentially limiting consistency in pain management practices across settings. Examining these conditions may provide a more comprehensive understanding of the determinants of effective pain management practices, inform the development of targeted interventions, and strengthen structural support systems that enable nurses to adhere to established protocols and deliver appropriate pain management practices.

## 4.1 Implementations and recommendations

In clinical practice, nurses need supported through structured and ongoing professional development programs to strengthen knowledge and skills in pain management practices. Healthcare facilities benefit from proving regular (i.e., annual or semi-annual) simulation-based and unit-based training to enhance pain management competencies and build nurses' confidence. In addition, establishing mentorship programs offer guidance, feedback, and role modeling; these further helps develop clinical skills and support the application of up-to-date evidence in practice. Pain-specific

hospital policies help to enforce the consistent use of both pharmacological and non-pharmacological pain management approaches.

Pain management education must be systematically integrated into nursing curricula, particularly during clinical training, to better prepare nursing students and future nurses for real-world practice.

Future research is needed to identify additional factors influencing pain management practices, especially system-level and organizational determinants. Further studies are also needed to validate and implement reliable pain assessment tools for older adults with cognitive impairment, a population in which pain is often underdiagnosed, underrecognized, and undertreated.

### 4.2 Limitations

This study has several limitations that should be acknowledged. First, the cross-sectional design was used, which could not demonstrate causality. Future longitudinal studies are needed to better explain the relationships among the variables. Second, the study was conducted in Thailand, where the nursing workforces is predominantly female (98.7%) [62]. Thus, the findings may not be generalized to other geographic regions or cultural contexts. Replicating of this study in diverse healthcare settings, including primary and secondary care hospitals and among male nurses, is recommended to confirm the findings. Finally, the use of self-report measures may introduce common method bias and increase the risk of social desirability bias. Future studies are encouraged to incorporate multi-source data collection (i.e., observational measures such as checklists) to minimize these potential biases.

### 5. Conclusion

This study underscores the importance of pain management self-efficacy in shaping clinical practices among Thai nurses caring for older adults with cognitive impairment. Targeted educational and training interventions at the individual, institutional, and system levels are needed to strengthen nurses' capability to deliver effective pain management. In addition, ongoing competency assessment is essential to ensure that nurses maintain up-to-date, evidence-based knowledge and skills to support improved pain management practices.

### Acknowledgments

The authors wish to thank all registered nurses who participated in this study. We would like to express our sincere gratitude to Research Assistant Professor Dr. Chang G. Park, Department of Population Health Nursing Science, Office of Research Facilitation, College of Nursing, University of Illinois at Chicago, as a consultant for data analysis.

### Author contributions

**Conceptualization:** Phichpraorn Youngcharoen, Atsadaporn Niyomyart, Piyawadee Thongyost, Joachim G. Voss.

**Data curation:** Phichpraorn Youngcharoen, Atsadaporn Niyomyart, Piyawadee Thongyost.

**Formal analysis:** Phichpraorn Youngcharoen, Atsadaporn Niyomyart, Piyawadee Thongyost, Joachim G. Voss.

**Funding acquisition:** Phichpraorn Youngcharoen.

**Investigation:** Phichpraorn Youngcharoen.

**Methodology:** Phichpraorn Youngcharoen, Atsadaporn Niyomyart, Piyawadee Thongyost, Joachim G. Voss.

**Project administration:** Phichpraorn Youngcharoen, Atsadaporn Niyomyart, Piyawadee Thongyost, Joachim G. Voss.

**Resources:** Phichpraorn Youngcharoen, Atsadaporn Niyomyart, Piyawadee Thongyost, Joachim G. Voss.

**Software:** Phichpraorn Youngcharoen, Atsadaporn Niyomyart, Piyawadee Thongyost, Joachim G. Voss.

**Supervision:** Phichpraorn Youngcharoen, Atsadaporn Niyomyart, Piyawadee Thongyost, Joachim G. Voss.

**Validation:** Phichpraorn Youngcharoen, Atsadaporn Niyomyart, Piyawadee Thongyost, Joachim G. Voss.

**Visualization:** Phichpraorn Youngcharoen, Atsadaporn Niyomyart, Piyawadee Thongyost, Joachim G. Voss.

**Writing – original draft:** Phichpraorn Youngcharoen, Atsadaporn Niyomyart, Piyawadee Thongyost, Joachim G. Voss.

**Writing – review & editing:** Phichpraorn Youngcharoen, Atsadaporn Niyomyart, Piyawadee Thongyost, Joachim G. Voss.

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
