## [Decision Letter · Decision Letter 0]

4 Jan 2026

PONE-D-25-58744Path analysis of Factors Associated with Nurses’ Pain Management Practice in Older Adults with Cognitive Impairment: a cross-sectional studyPLOS One

Dear Dr. Niyomyart,

Thank you for submitting your manuscript to PLOS ONE. After careful consideration, we feel that it has merit but does not fully meet PLOS ONE’s publication criteria as it currently stands. Therefore, we invite you to submit a revised version of the manuscript that addresses the points raised during the review process.

We look forward to receiving your revised manuscript.

Kind regards,

Ahmed Abdelwahab Ibrahim El-Sayed

Academic Editor

PLOS One

**Journal Requirements:**

1. When submitting your revision, we need you to address these additional requirements. Please ensure that your manuscript meets PLOS ONE's style requirements, including those for file naming. The PLOS ONE style templates can be found at https://journals.plos.org/plosone/s/file?id=wjVg/PLOSOne_formatting_sample_main_body.pdf and https://journals.plos.org/plosone/s/file?id=ba62/PLOSOne_formatting_sample_title_authors_affiliations.pdf 2. Please provide additional details regarding participant consent. In the ethics statement in the Methods and online submission information, please ensure that you have specified what type you obtained (for instance, written or verbal, and if verbal, how it was documented and witnessed). If your study included minors, state whether you obtained consent from parents or guardians. If the need for consent was waived by the ethics committee, please include this information. Once you have amended this/these statement(s) in the Methods section of the manuscript, please add the same text to the “Ethics Statement” field of the submission form (via “Edit Submission”). For additional information about PLOS ONE ethical requirements for human subjects research, please refer to http://journals.plos.org/plosone/s/submission-guidelines#loc-human-subjects-research. 3. Thank you for stating the following financial disclosure: Ramathibodi School of Nursing, Faculty of Medicine Ramathibodi Hospital, Mahidol University, will support the publication fee.    Please state what role the funders took in the study.  If the funders had no role, please state: "The funders had no role in study design, data collection and analysis, decision to publish, or preparation of the manuscript." If this statement is not correct you must amend it as needed. Please include this amended Role of Funder statement in your cover letter; we will change the online submission form on your behalf. 4. In the online submission form, you indicated that “Data are available upon request.” All PLOS journals now require all data underlying the findings described in their manuscript to be freely available to other researchers, either a. In a public repository, b. Within the manuscript itself, or c. Uploaded as supplementary information.This policy applies to all data except where public deposition would breach compliance with the protocol approved by your research ethics board. If your data cannot be made publicly available for ethical or legal reasons (e.g., public availability would compromise patient privacy), please explain your reasons on resubmission and your exemption request will be escalated for approval. 5. If the reviewer comments include a recommendation to cite specific previously published works, please review and evaluate these publications to determine whether they are relevant and should be cited. There is no requirement to cite these works unless the editor has indicated otherwise.

**Additional Editor Comments**

Dear Authors,

Thank you for your submission to PLOS ONE. The reviewers have identified several important points and raised a number of issues that need to be carefully addressed before we can consider your manuscript further.

We look forward to receiving your revised manuscript.

Reviewers' comments:

Reviewer's Responses to Questions

**Comments to the Author**

1. Is the manuscript technically sound, and do the data support the conclusions?

Reviewer #1: Yes

Reviewer #2: Yes

Reviewer #3: Yes

Reviewer #4: Yes

2. Has the statistical analysis been performed appropriately and rigorously? 

Reviewer #1: Yes

Reviewer #2: Yes

Reviewer #3: Yes

Reviewer #4: Yes

3. Have the authors made all data underlying the findings in their manuscript fully available?

Reviewer #1: Yes

Reviewer #2: Yes

Reviewer #3: Yes

Reviewer #4: No

4. Is the manuscript presented in an intelligible fashion and written in standard English?

Reviewer #1: Yes

Reviewer #2: Yes

Reviewer #3: Yes

Reviewer #4: Yes

5. Review Comments to the Author

**Reviewer #1:** Overall significance

• The manuscript examines a significant and underexplored domain: the pain management practices of nurses concerning older adults with cognitive impairment, employing a theoretically informed path analysis. The integration of Social Cognitive Theory and the analysis of self-efficacy as a mediator significantly contributes to nursing and gerontological pain literature, especially in the Thai context.

• The manuscript requires professional English language editing.

• The references are up-to-date and appropriate for publication.

• Clarity and Structure: The manuscript is well organized and clearly articulated. Certain sections would benefit from enhanced conciseness.

Abstract

It is advisable to briefly indicate the cross-sectional and secondary analysis characteristics of the study in the Methods section to improve transparency.

Background

• The background offers a compelling rationale, underpinned by pertinent and current literature.

• The transition from prevalence to clinical consequences to nursing challenges is coherent.

Suggestions:

• Some paragraphs (pages 11–12) are dense and could be streamlined by reducing overlapping citations.

• Clearly identify the existing knowledge gap by noting that previous path analyses or mediation models have not investigated these variables in older adults with cognitive impairments.

• Explicitly reference and interpret Figure 1 in relation to the hypothesized paths.

Methods

• Study Design: Add a brief justification for why secondary analysis was suitable for addressing the current research aim.

• Tools: Clarify how adding new items to TENAP Part B may affect comparability with prior studies.

• Data Analysis: The statistical approach is appropriate and well justified.

• Briefly state how assumptions for SEM (normality, multicollinearity) were assessed.

Results:

• The results are presented clearly and sequenced logically.

Discussion:

• The discussion is clearly articulated, and the interpretation of self-efficacy as a mediator is notably robust.

• Reduce redundancy in the discussion of knowledge, collaboration, and experience throughout various paragraphs.

• To improve international relevance, explicitly compare findings with non-Asian or Western contexts.

Limitations:

• Add a brief note on common method bias due to self-report measures.

Conclusion:

• Refrain from repeating detailed findings; rather, highlight the significance of noting essential points.

**Reviewer #2:** Objective does not state clearly what is the goal of study?, what research will achieve/outcome after finding relationships. it is not clear. the study needs more strong need assessment as to what happen due to limited knowledge of pain management in terms of cognitive impairment elderly clients?, Discuss factors behind direct and indirect relation among variables and impacts in quantifiable data, more studies from different countries is required in discussion. adding the impact is crucial in discussion. recommendation is required in separate heading, the topic is vital to understand and explore more on pain management, appreciated to develop the research. the study can also explore views from patient their perceptions to relate further the impact.

**Reviewer #3:** This study focuses on nurses' pain management practices in this population, which closely aligns with clinical needs and holds practical relevance. However, the following areas still require improvement:

1.Inconsistent use of terminology: For example, “collaborative perception” and “perception of collaboration” are used interchangeably. It is recommended to unify the terms throughout the manuscript.

2. The model explained only 37% of the variance. It is suggested that the discussion briefly addresses other potential factors influencing pain management practices, such as organizational culture and resource support.

3. The findings support that improving nurses’ self-efficacy can enhance pain management practices. It is recommended to explicitly propose targeted training strategies in the “Conclusion” or “Recommendations” section, such as scenario-based simulation and reflective practice.

4. All participants were female. It is suggested that this limitation be further explained in the discussion, and future studies should consider including male nurses or nursing populations with different educational backgrounds.

**Reviewer #4:** This manuscript addresses an important and underexplored topic: factors associated with nurses’ pain management practices for older adults with cognitive impairment. The topic is clinically relevant, theoretically grounded, and aligned with PLOS ONE’s scope. The use of Social Cognitive Theory and path analysis provides a coherent conceptual framework, and the sample size appears adequate for the proposed analyses. Overall, the study has merit; however, several issues should be addressed to improve clarity, rigor, and transparency before the manuscript can be considered for publication.

Major comments

The Introduction would benefit from structural consolidation. Currently, background information, literature review, and conceptual framing are distributed across multiple sections with some repetition. These sections could be merged into a single, more concise Introduction that progresses from (1) the clinical importance of pain management in older adults with cognitive impairment, to (2) existing evidence on nurses’ knowledge, attitudes, collaboration, and experience, and (3) the theoretical grounding in Social Cognitive Theory leading to the study aim. This restructuring would improve flow, reduce redundancy, and enhance readability without loss of content.

Conceptual justification of the model

The manuscript states that Social Cognitive Theory (SCT) guided the model; however, only a subset of SCT constructs (knowledge/attitudes, collaboration, self-efficacy) are operationalized. Other core SCT components such as outcome expectations, observational learning, and reinforcement are not included. The authors should strengthen the theoretical justification explaining why the selected constructs were prioritized and how collaboration and years of experience conceptually map onto SCT domains. This clarification would strengthen the conceptual coherence of the model.

Model modification and path removal

The manuscript notes that the initial model was just-identified and that the path from years of experience to pain management practice was removed to improve model fit. This modification requires clearer justification. Please clarify whether this decision was theory-driven or guided by model diagnostics (e.g., modification indices). Post-hoc model changes should be explicitly justified to avoid concerns about data-driven model fitting.

Interpretation of mediation using cross-sectional data

Although bootstrapped mediation was conducted appropriately, the cross-sectional design limits causal inference. Throughout the Results and Discussion, causal wording such as “influences,” “improves,” or “leads to” should be softened and replaced with language such as “is associated with” or “is consistent with an indirect relationship.” This clarification is important to avoid overstating causal conclusions.

Measurement considerations and self-report bias

All key variables, including pain management practice, were measured using self-report instruments. This raises the possibility of social desirability and common-method bias. While this limitation is acknowledged, it should be more explicitly emphasized. The authors are encouraged to discuss how future studies could incorporate observational measures, audits, or objective indicators of practice.

Modification of the TENAP instrument

The manuscript describes substantial adaptation of the TENAP instrument, including adding new items and expanding its scope from assessment to management. While psychometric testing is reported, the authors should more clearly acknowledge that this represents an expanded or modified version of the original instrument. Clarification is needed regarding construct equivalence and comparability with prior TENAP-based studies.

Statistical reporting clarity

Please clarify whether coefficients reported in tables and figures are standardized or unstandardized.

Ensure consistency in reporting correlation coefficients (Pearson vs. Spearman) and notation.

Some confidence intervals appear wide relative to the reported coefficients and should be double-checked.

Minor formatting inconsistencies (spacing, p-value notation) should be corrected.

Generalizability

The study sample consists entirely of female nurses from two hospitals in Thailand. While this reflects the local workforce context, the authors should more clearly acknowledge limits to generalizability beyond similar cultural and healthcare settings.

Minor comments

-Minor grammatical and stylistic issues are present throughout the manuscript and should be corrected during revision.

-Keywords should be standardized (e.g., “nurses’ pain management” rather than “nurse’ pain management”).

-Figures would benefit from clearer labeling and explicit indication of standardized coefficients.

-Some repetition in the Discussion could be reduced to improve clarity and conciseness.

Summary

This study addresses a relevant clinical and educational issue and applies an appropriate analytical approach. With clearer theoretical justification, more cautious interpretation of mediation effects, improved reporting transparency, and minor editorial revisions, the manuscript would be suitable for publication.

6. PLOS authors have the option to publish the peer review history of their article (what does this mean?). If published, this will include your full peer review and any attached files.

Reviewer #1: No

Reviewer #2: **Yes:** Zaibunissa

Reviewer #3: No

Reviewer #4: No

---

## [Author Response · Author response to Decision Letter 1]

9 Mar 2026

Path analysis of Factors Associated with Nurses’ Pain Management Practice in Older Adults with Cognitive Impairment: a cross-sectional study

Editor Comments Author’s response

Overall significance

• The manuscript examines a significant and underexplored domain: the pain management practices of nurses concerning older adults with cognitive impairment, employing a theoretically informed path analysis. The integration of Social Cognitive Theory and the analysis of self-efficacy as a mediator significantly contributes to nursing and gerontological pain literature, especially in the Thai context.

• The manuscript requires professional English language editing.

• The references are up-to-date and appropriate for publication.

• Clarity and Structure: The manuscript is well organized and clearly articulated. Certain sections would benefit from enhanced conciseness. Thank you so much for the positive and overall evaluation of our manuscript. Regarding the English language quality, the manuscript has been edited by a native speaker to improve clarity and readability throughout according to your comment.

Abstract

It is advisable to briefly indicate the cross-sectional and secondary analysis characteristics of the study in the Methods section to improve transparency. Thank you so much for your feedback. We have added a justification for the use of secondary data analysis as follows:

Study Design: The secondary analysis was aligned with the study objectives and the original data collection purpose and was supported by validated measurement tools and an adequate sample size.

A justification has been added to both the abstract and the main text.

Background

• The background offers a compelling rationale, underpinned by pertinent and current literature.

• The transition from prevalence to clinical consequences to nursing challenges is coherent. Thank you so much for your positive feedback.

Suggestions:

• Some paragraphs (pages 11–12) are dense and could be streamlined by reducing overlapping citations.

• Clearly identify the existing knowledge gap by noting that previous path analyses or mediation models have not investigated these variables in older adults with cognitive impairments.

• Explicitly reference and interpret Figure 1 in relation to the hypothesized paths. Thank you for your insightful suggestion!

We have reduced overlapping citations and prioritized recent references throughout the background section. However, Bandura’s foundational theory of self-efficacy was retained, as it is essential to the conceptual framework. These revisions improve clarity and coherence while preserving theoretical integrity.

Methods

• Study Design: Add a brief justification for why secondary analysis was suitable for addressing the current research aim.

• Tools: Clarify how adding new items to TENAP Part B may affect comparability with prior studies.

• Data Analysis: The statistical approach is appropriate and well justified.

• Briefly state how assumptions for SEM (normality, multicollinearity) were assessed.

Results:

• The results are presented clearly and sequenced logically. Thank you so much for your feedback. We have added a justification for the use of secondary data analysis as follows:

Study Design: The secondary analysis was aligned with the study objectives and the original data collection purpose and was supported by validated measurement tools and an adequate sample size.

A justification has been added to both the abstract and the main text.

Tools: We add a clarification on page 6, second paragraph as follows:

As no existing instrument specifically evaluates pain management in older adults with cognitive impairment, we reviewed the literature and obtained approval from the developer to expand the instrument. Thus, eleven additional items were added to each vignette to assess nurses’ pain management practices in this population. Following the development, the Thai questions were submitted to five Thai experts for feedback on clarity, relevance, and logical flow.

Data analysis:

Thank you for your suggestion. We have included how SEM assumptions were assessed as follows:

Page 8, paragraph 2 under Data analysis:

Variance inflation factors (VIF), skewness, and kurtosis were assessed to evaluate potential violations of multicollinearity and normality assumptions. In this study, all VIF values were below 10, indicating no evidence of problematic multicollinearity among the independent variables. However, the distributions of years of nursing experience and nurses’ pain management practice scores demonstrated notable skewness, suggesting deviation from normality.

Page 8, paragraph 4: Path analysis was conducted using structural equation modeling (SEM). To estimate the coefficients of direct effects and to test mediation effects, we employed a bootstrapping method with 5,000 resamples; bootstrapping is recommended for evaluating the significance of indirect effects as it does not rely on the assumption of normality of the sampling distribution and provides more accurate confidence intervals, therefore reducing the risk of Type I error (36).

Results: Thank you so much for your feedback.

Discussion:

• The discussion is clearly articulated, and the interpretation of self-efficacy as a mediator is notably robust.

• Reduce redundancy in the discussion of knowledge, collaboration, and experience throughout various paragraphs.

• To improve international relevance, explicitly compare findings with non-Asian or Western contexts. Thank you so much for your feedback. We reviewed the entire manuscript and minimized redundancy throughout. However, as research on pain management among older adults with cognitive impairment remains limited, we were only able to identify a small number of comparable studies conducted in Asian populations. We hope the current comparisons are information and appropriate to be considered

Limitations:

• Add a brief note on common method bias due to self-report measures. Thank you for your comment. We acknowledge that the use of self-report questionnaires may introduce common method bias, which cannot be fully eliminated and should be carefully considered when interpreting the findings. To minimize this issue in our study, we assured anonymity and confidentiality of participants as well as used clear and neutral wording throughout the questionnaire. A statement below has now been added to the limitation section.

Moreover, common method bias due to self-report measures may increase the risk of social desirability bias. Therefore, future studies are encouraged to use multi-source data collection (i.e., observational measures such as checklists) to minimize its potential impact.

Conclusion:

• Refrain from repeating detailed findings; rather, highlight the significance of noting essential points. Thank you so much for your valuable suggestion. We have revised our conclusion as follows:

The findings underscore the importance of nurses’ knowledge and attitudes, collaborative perceptions, years of experience, and pain management self-efficacy in shaping pain management practices. This information highlights the need for targeted interventions and policy development to improve pain management for older adults with cognitive impairment, a vulnerable population in Thailand.

Reviewer 2 Comments Author’s response

Objective does not state clearly what is the goal of study?, what research will achieve/outcome after finding relationships. it is not clear. the study needs more strong need assessment as to what happen due to limited knowledge of pain management in terms of cognitive impairment elderly clients?, Discuss factors behind direct and indirect relation among variables and impacts in quantifiable data, more studies from different countries is required in discussion. adding the impact is crucial in discussion. recommendation is required in separate heading, the topic is vital to understand and explore more on pain management, appreciated to develop the research. the study can also explore views from patient their perceptions to relate further the impact. Thank you so much for your feedback. We have revised the objective and finding outcome to make it clearer as follows:

Background: Pain management is a fundamental component of patient care. Inadequate pain management in patients is associated with increased anxiety, depression, and reduced quality of life. Several studies assessed the nurse’ pain management practice in Thailand; however, research focusing on older adults with cognitive impairment remains limited. Therefore, the study aimed to investigate relationships among nurses’ knowledge and attitude, collaborative perception, years of experience, pain management self-efficacy, and their pain management practices.

Conclusions: The findings…... Thus, targeted intervention, structured in-service training programs, and supportive policy development should be initiated to enhance pain management competencies and improve the quality of nursing care among older adults with cognitive impairment.

Reviewer 3 Comments Author’s response

This study focuses on nurses' pain management practices in this population, which closely aligns with clinical needs and holds practical relevance. However, the following areas still require improvement: Thank you so much for your feedback.

1.Inconsistent use of terminology: For example, “collaborative perception” and “perception of collaboration” are used interchangeably. It is recommended to unify the terms throughout the manuscript. Thank you so much for your valuable comment. After carefully reviewing the manuscript, we have revised the terminology throughout to consistently use the term “perception of collaboration” to ensure clarity and alignment with the measurement tool, as this terminology reflects how nurses perceive their collaboration with physicians.

2. The model explained only 37% of the variance. It is suggested that the discussion briefly addresses other potential factors influencing pain management practices, such as organizational culture and resource support. Thank you so much for your comment. We have added further clarification and supporting information as follows:

Page 17, last paragraph:

Pain management is a fundamental component of effective nursing care, serving as a nursing-sensitive indicator and a continuous nursing-focused process that reflects the quality of pain interventions (45, 46). However, pain management practices are complex and associated with multiple factors. While this study identified several significant individual-level factors, a substantial proportion of variance in pain management practices remained unexplained. Prior studies have shown that pain management is related not only to individual characteristics but also to organizational factors (47, 48). Therefore, systemic conditions within the Thai healthcare context warrant further investigation. Such efforts may inform the development of hospital-level policies and ultimately enhance nurses’ competencies in pain management.

3. The findings support that improving nurses’ self-efficacy can enhance pain management practices. It is recommended to explicitly propose targeted training strategies in the “Conclusion” or “Recommendations” section, such as scenario-based simulation and reflective practice. Thank you for your suggestion. We have revised our conclusion as follows:

5. Conclusion

The findings underscore the importance of pain management self-efficacy in relation to pain management practices. These results highlight the need for targeted interventions and training strategies such as scenario-based simulation and reflective practice along with supportive institutional policies, may help strengthen pain management for older adults with cognitive impairment.

4. All participants were female. It is suggested that this limitation be further explained in the discussion, and future studies should consider including male nurses or nursing populations with different educational backgrounds. Thank you for your comment. We have now updated the limitations sections as follows:

4.1 Limitations

This study has several limitations that should be acknowledged. First, a cross-sectional design was used, which could not demonstrate causality. Future longitudinal studies are needed to better explain the relationships among the variables. Second, the study was conducted in Thailand, where 98.7% of the nursing workforce is female (49). Thus, the findings may not be generalized to other geographic regions and cultural contexts. Replicating of this study in primary and secondary care hospitals mor among male nurses is recommended to confirm the findings. Finally, the use of self-report measures may introduce common method bias and increase the risk of social desirability bias. Future studies are encouraged to incorporate multi-source data collection (i.e., observational measures such as checklists) to minimize this potential impact.

Reviewer 4 Comments Author’s response

This manuscript addresses an important and underexplored topic: factors associated with nurses’ pain management practices for older adults with cognitive impairment. The topic is clinically relevant, theoretically grounded, and aligned with PLOS ONE’s scope. The use of Social Cognitive Theory and path analysis provides a coherent conceptual framework, and the sample size appears adequate for the proposed analyses. Overall, the study has merit; however, several issues should be addressed to improve clarity, rigor, and transparency before the manuscript can be considered for publication. Thank you so much for your kindly feedback.

Major comments

The Introduction would benefit from structural consolidation. Currently, background information, literature review, and conceptual framing are distributed across multiple sections with some repetition. These sections could be merged into a single, more concise Introduction that progresses from (1) the clinical importance of pain management in older adults with cognitive impairment, to (2) existing evidence on nurses’ knowledge, attitudes, collaboration, and experience, and (3) the theoretical grounding in Social Cognitive Theory leading to the study aim. This restructuring would improve flow, reduce redundancy, and enhance readability without loss of content. Thank you for your suggestion. We have revised and reorganized the Introduction to improve flow and eliminate repetition in accordance with your feedback. This restructuring now enhances coherence and readability without losing content.

Conceptual justification of the model

The manuscript states that Social Cognitive Theory (SCT) guided the model; however, only a subset of SCT constructs (knowledge/attitudes, collaboration, self-efficacy) are operationalized. Other core SCT components such as outcome expectations, observational learning, and reinforcement are not included. The authors should strengthen the theoretical justification explaining why the selected constructs were prioritized and how collaboration and years of experience conceptually map onto SCT domains. This clarification would strengthen the conceptual coherence of the model. Thank you so much for your feedback. Our study focused on components that are directly related to nurses’ pain management practices in clinical settings; therefore, only theoretically relevant components were included. Below our revisions:

1.1 Conceptual Framework

This study was guided by Social Cognitive Theory (SCT) and relevant literature review. Social Cognitive Theory, developed by Albert Bandura, describes how personal, behavioral, and environmental factors interact to influence individual behaviors (19). Although SCT includes multiple key components (i.e., self-efficacy, outcome expectations, observational learning, and reinforcement), this study focused on those most directly related to nurses’ pain management practices in clinical settings (19). Thus, only theoretically relevant components were operationalized.

Self-efficacy, one of the key elements of SCT, refers to an individual’s belief in their ability to perform specific tasks, influencing behavior change or practice (19). According to Bandura, self-ef

---

## [Decision Letter · Decision Letter 1]

31 Mar 2026

PONE-D-25-58744R1Path analysis of Factors Associated with Nurses’ Pain Management Practice in Older Adults with Cognitive Impairment: a cross-sectional studyPLOS One

Dear Dr. Niyomyart,

Thank you for submitting your manuscript to PLOS ONE. After careful consideration, we feel that it has merit but does not fully meet PLOS ONE’s publication criteria as it currently stands. Therefore, we invite you to submit a revised version of the manuscript that addresses the points raised during the review process.

We look forward to receiving your revised manuscript.

Kind regards,

Ahmed Abdelwahab Ibrahim El-Sayed

Academic Editor

PLOS One

Journal Requirements:

Additional Editor Comments

Dear Authors,

Thank you for your revision. Your manuscript has now been carefully assessed. However, several issues have been raised by the reviewer during this round, which need to be addressed carefully before we can consider your paper further

Reviewers' comments:

Reviewer's Responses to Questions

**Comments to the Author**

1. If the authors have adequately addressed your comments raised in a previous round of review and you feel that this manuscript is now acceptable for publication, you may indicate that here to bypass the “Comments to the Author” section, enter your conflict of interest statement in the “Confidential to Editor” section, and submit your "Accept" recommendation.

Reviewer #1: All comments have been addressed

2. Is the manuscript technically sound, and do the data support the conclusions?

Reviewer #1: Partly

3. Has the statistical analysis been performed appropriately and rigorously? 

Reviewer #1: Yes

4. Have the authors made all data underlying the findings in their manuscript fully available?

Reviewer #1: No

5. Is the manuscript presented in an intelligible fashion and written in standard English?

Reviewer #1: No

6. Review Comments to the Author

Reviewer #1: Abstract: Keywords are not fully aligned with MeSH terminology.

Literature Review and Conceptual Framework: Figure 1 is not sufficiently explained or integrated into the text.

Methods

· Participant flow, inclusion/exclusion criteria, and response rate are not clearly reported.

· No a priori sample size calculation or statistical power analysis is clearly justified.

Measurement Validity and Reliability

· Scoring procedures and interpretation of scale direction are inconsistently described.

· The modification of the TENAP instrument is insufficiently justified in terms of construct validity.

· The comparability with previous studies using the original instrument is not adequately addressed.

· The manuscript does not report the presence, proportion, or handling of missing data.

Results

· Descriptive statistics lack consistent formatting across tables.

· Tables do not consistently include footnotes explaining scale ranges and score interpretation.

· Percentages are inconsistently reported (decimal places vary).

· Table titles do not consistently report sample size (N).

· Variable labels are inconsistently formatted.

· Use of commas instead of decimal points violates international reporting standards.

· Structural model results are presented clearly; however, confidence intervals and effect sizes are not fully integrated into the narrative.

· The non-significant paths are not adequately contextualized.

· The explained variance (37%) is not critically interpreted.

Discussion

· Direct and indirect relationships are not sufficiently explained mechanistically.

· Limited comparison with international (non-Asian) studies reduces global relevance.

· Some statements approach causal interpretation despite a cross-sectional design.

· Residual redundancy is present across discussion sections.

Implications and Recommendations

· A separate “Recommendations” section is not provided despite the reviewer’s request.

· Practical implications remain general and lack actionable strategies.

· Clinical, educational, and policy-level implications are not clearly differentiated.

References

· The reference list does not consistently follow the journal’s required citation style.

· The manuscript relies heavily on region-specific sources, limiting global applicability.

7. PLOS authors have the option to publish the peer review history of their article (what does this mean?). If published, this will include your full peer review and any attached files.

Reviewer #1: No

---

## [Author Response · Author response to Decision Letter 2]

1 May 2026

Abstract: Keywords are not fully aligned with MeSH terminology

Response: Thank you for your valuable feedback. We have revised the keywords to ensure alignment with MeSH terminology. The updated keywords are: Cognitive dysfunction; collaboration; nurses; knowledge; pain management; self-efficacy

Literature Review and Conceptual Framework: Figure 1 is not sufficiently explained or integrated into the text.

Response: Thank you for the comment. We have revised the manuscript to better explain and integrate Figure 1 into the text, including clarification of the conceptual framework and hypothesized relationships. These revisions have been incorporated on page 4.

Methods

- Participant flow, inclusion/exclusion criteria, and response rate are not clearly reported.

- No a priori sample size calculation or statistical power analysis is clearly justified.

Response: Thank you so much for your feedback. We have revised the Methods section to clarify the participant flow, inclusion and exclusion criteria, and response rate. In addition, a prior sample size calculation and statistical power analysis have been included. These revisions are presented on page 5.

Measurement Validity and Reliability

1. Scoring procedures and interpretation of scale direction are inconsistently described.

2. The modification of the TENAP instrument is insufficiently justified in terms of construct validity.

3. The comparability with previous studies using the original instrument is not adequately addressed.

4. The manuscript does not report the presence, proportion, or handling of missing data.

Response:

1. We have revised the Methods section to clarify the scoring procedures and ensure consistency in the interpretation of scale direction for the TENAP (pages 6-7) and CSACD (page 8). Specifically, we clarified item scoring, Likert scale ranges, and the interpretation to total scores.

2. We have revised the manuscript to strengthen the justification for the modification of the TENAP instrument by providing additional evidence of construct validity. We examined convergent and divergent validity, showing a moderate positive correlation with nurses’ pain management self-efficacy and no significant correlation with work complexity. These revisions have been added on page 8.

3. We have revised the manuscript to clearly describe the differences between the original and modified TENAP and to clarify the implication for comparability. We added a paragraph explaining that the original TENAP assessed nurses’ knowledge and attitudes toward pain management and pain assessment, whereas the modified version expands the instrument to include pain management practices, resulting in both conceptual and structural differences (additional items and an expanded scoring range). We also clarified that these differences may limit direct comparability with previous-based studies. These revisions have been added on page 7.

4. We reported how we handled missing data on pages 5 and 6. Specifically, questionnaires were checked for completeness upon return, and participants were given the option to complete any accidentally missed items or submit the questionnaire without modification. Initial screening identified minimal missing data, and all participants chose to complete the missing items, resulting in a final dataset with no missing data.

Results

1. Descriptive statistics lack consistent formatting across tables.

2. Tables do not consistently include footnotes explaining scale ranges and score interpretation.

3. Percentages are inconsistently reported (decimal places vary).

4. Table titles do not consistently report sample size (N).

5. Variable labels are inconsistently formatted.

6. Use of commas instead of decimal points violates international reporting standards.

7. Structural model results are presented clearly; however, confidence intervals and effect sizes are not fully integrated into the narrative.

8. The non-significant paths are not adequately contextualized.

9. The explained variance (37%) is not critically interpreted.

Response: Thank you for these helpful suggestions. We have revised all tables to address the issues raised (points 1–5). Specifically, we standardized the formatting of descriptive statistics, added footnotes explaining scale ranges and score interpretation, ensured consistent reporting of percentages (uniform decimal places), included sample sizes (N) in all table titles, and revised variable labels for consistency throughout the manuscript.

6. We have revised all numerical values throughout the manuscript to ensure the use of decimal points according to international reporting standards.

7. We have revised the results section to fully integrate confidence intervals and effect sizes into the narrative description of the structural model findings. In addition, confidence interval values have been corrected and are now constituently reported alongside effect sizes to provide a complete interpretation of the results. The revision has been made on pages 12-13.

8. We have revised the Results section to better contextualize the non-significant paths by providing an interpretation of their implications on page 12.

9. We have revised the manuscript to better interpret the explained variance, as follows: The final model accounted for 37% of the variance in nurses’ pain management practices (R2 = 0.37), indicating that the model explains a moderate amount of variance. The remaining 63% suggests that additional factors not included in the model may influence pain management practices.

Thank you so much for your feedback.

Discussion

1. Direct and indirect relationships are not sufficiently explained mechanistically.

2. Limited comparison with international (non-Asian) studies reduces global relevance.

3. Some statements approach causal interpretation despite a cross-sectional design.

4. Residual redundancy is present across discussion sections.

Response: We have revised the Discussion section to improve clarify and interpretation across the section. We added more comparisons with international studies to enhance global relevance. We also revised wording throughout to avoid causal interpretations. Also, redundant statements were reduced to improve clarity and conciseness.

Implications and Recommendations

1. A separate “Recommendations” section is not provided despite the reviewer’s request.

2. Practical implications remain general and lack actionable strategies.

3. Clinical, educational, and policy-level implications are not clearly differentiated.

Response: Thank you for your valuable suggestion. We have added a separate “Implications and Recommendations” section and revised the content to provide more specific and actionable strategies across clinical, educational, and policy levels in response to your feedback.

References

1. The reference list does not consistently follow the journal’s required citation style.

2. The manuscript relies heavily on region-specific sources, limiting global applicability.

Response: Thank you for these helpful comments. We have reviewed and revised all references to ensure consistency with the journal’s citation style. Also, we have included additional international studies from diverse regions to enhance the global relevance and strengthen the comparative context of the manuscript.

Note: We have also revised the conclusion sections in both the abstract and the main text to ensure alignment.

---

## [Editor Report · Decision Letter 2]

8 May 2026

Path analysis of Factors Associated with Nurses’ Pain Management Practices in Older Adults with Cognitive Impairment: a cross-sectional study

PONE-D-25-58744R2

Dear Author,

We’re pleased to inform you that your manuscript has been judged scientifically suitable for publication and will be formally accepted for publication once it meets all outstanding technical requirements.

Kind regards,

Ahmed Abdelwahab Ibrahim El-Sayed,

Academic Editor

PLOS One

Additional Editor Comments (optional):

Dear Authors,

Thank you for your revision. I have finished my review to your revision, and I can accept your manuscript for publication in its current form at PLOS ONE.

---

## [Editor Report · Acceptance letter]

PONE-D-25-58744R2

PLOS One

Dear Dr. Niyomyart,

I'm pleased to inform you that your manuscript has been deemed suitable for publication in PLOS One. Congratulations! Your manuscript is now being handed over to our production team.

Kind regards,

on behalf of

Dr. Ahmed Abdelwahab Ibrahim El-Sayed

Academic Editor

PLOS One